# Evaluation of Poly Lactic-co-Glycolic Acid-Coated β-Tricalcium Phosphate Bone Substitute as a Graft Material for Ridge Preservation after Tooth Extraction in Dog Mandible: A Comparative Study with Conventional β-Tricalcium Phosphate Granules

**DOI:** 10.3390/ma13163452

**Published:** 2020-08-05

**Authors:** Takamitsu Koga, Shinsuke Kumazawa, Yusuke Okimura, Yumi Zaitsu, Kazuhiko Umeshita, Izumi Asahina

**Affiliations:** 1Department of Regenerative Oral Surgery, Institute of Biomedical Sciences, Nagasaki University, Nagasaki 852-8588, Japan; kogataka215@gmail.com; 2Department of Dentistry and Oral Surgery, Imaki-ire General Hospital, Kagoshima 892-8502, Japan; 3R&D Department, Sunstar Inc., Takatsuki 569-1195, Japan; shinsuke.kumazawa@jp.sunstar.com (S.K.); yusuke.okimura@jp.sunstar.com (Y.O.); yumi.zaitsu@jp.sunstar.com (Y.Z.); 4Safety Analysis R&D Promotion Department, Sunstar Inc., Takatsuki 569-1195, Japan; Kazuhiko.Umeshita@jp.sunstar.com

**Keywords:** bone substitute, β-tricalcium phosphate, poly lactic-co-glycolic acid, alveolar ridge preservation

## Abstract

This study aims to evaluate the safety and efficacy of a poly lactic-co-glycolic acid (PLGA)-coated β-tricalcium phosphate (β-TCP) with N-methyl-2-pyrrolidone (NMP) liquid activator (PLGA/β-TCP) on alveolar ridge preservation after tooth extraction in dog mandible. Thirty-two extraction sites were prepared in eight dog mandibles. A distal root of the mandibular premolar was extracted and randomly grafted with one of the following bone substitutes: (1) PLGA/β-TCP, (2) β-TCP, or (3) left empty as a control, and wounds were closed with keratinized mucosa graft. Post-operative wound healing was observed and scored to evaluate safety. After 12 and 24 weeks, the bone regeneration was evaluated with micro-computed tomography (CT) images and histomorphometric analyses. Gingival epithelization progressed over time without complication or infection. Micro-CT images and histological observation revealed that both PLGA/β-TCP and β-TCP granules supported sufficient new bone formation. Although bone formation and substrate resorption were delayed slightly with the PLGA and the NMP-containing plasticizer as compared to those treated with conventional β-TCP, it can be concluded that the PLGA and the NMP-containing plasticizer that facilitated the in situ hardening properties of the material had no negative influence on the biocompatibility of the material.

## 1. Introduction

The recovery of bone deficiency caused by trauma, tumor resection, and also aging has been a challenge in the field of orthopedics and also dentistry. The autogenous bone graft is thought to be a gold standard because of its superior osteogenesis; however, it has impediments such as limited availability and a donor site morbidity. In order to compensate these drawbacks, a variety of artificial bone graft materials has been developed and clinically applied for bone augmentation, and the materials selected significantly affect the outcome of bone replacement procedures in terms of bone formation volume and the quality and amount of vital bone [1].

Post-extraction alveolar ridge bone volume loss is an irreversible process involving both horizontal and vertical reduction [2,3]. Alveolar ridge atrophy may have a considerable impact on tooth replacement therapy, particularly for implant-supported restorations [4]. Therefore, alveolar ridge preservation has become a key component of contemporary clinical dentistry.

Various bone grafting products are used for bone augmentation in implant dentistry, including allograft, xenografts, and alloplastic bone grafts. The development of alloplastic bone grafts has been particularly notable. Non-absorbable grafting material, such as hydroxyapatite and glass ionomer cements, and absorbable grafting material, such as β-tricalcium phosphate (β-TCP), α-tricalcium phosphate, and carbonate apatite, are in clinical use. Effective bone grafting materials are required to be osteoconductive, biocompatible, and cost-effective. Furthermore, excellent operability and ease of maintenance in the local site are demanded for dental clinicians.

In recent years, several bone substitutes based on calcium phosphates, combined with different natural-based polymers, such as cellulose and its derivatives, hyaluronic acid, and other polymers, have been used as medical devices [5,6]. These materials fulfill the requirements of injectability, filling complex-shaped bone defects and setting very firmly in situ with osteoconductivity and degradability [7]. Among them, poly lactic-co-glycolic acid (PLGA)-coated β-TCP with N-methyl-2-pyrrolidone (NMP) liquid activator (PLGA/β-TCP) (GUIDOR *easy-graft* CLASSIC, granule size 500–1000 µm, Sunstar Suisse SA, Etoy, Switzerland) is a synthetic bone grafting system and a bone grafting material that hardens into a stable, porous scaffold within minutes and, therefore, may reduce the need for a barrier membrane to retain graft materials. This product comprises β-TCP granules coated with PLGA, a biodegradable polymer. Before application, this composite graft is mixed with the NMP liquid activator, which softens the PLGA, making the surface sticky and rendering the particles easy to mold. On contact with blood or other fluids, the liquid activator is excreted, and the particles form a solid scaffold in situ. The suitability of this material for ridge preservation has been clinically reported [1,8,9,10,11,12]. However, the impact of polymers on the resorption/replacement of PLGA/β-TCP and the negative effect of PLGA and NMP during wound healing of a tooth extraction socket are currently unknown, though there have been no negative effects reported in maxillary sinus floor augmentation both in an animal and a clinical study [13,14].

The purpose of this study was to evaluate the safety and efficacy of a PLGA/β-TCP bone substitute in preserving the alveolar ridge after tooth extraction in dog mandible.

## 2. Materials and Methods

This study was approved by the Institutional Animal Care and Use Committee of NISSEI BILIS Co., Ltd., Shiga laboratory (Permission number: 1812-14) and was conducted from 25 December 2018, to 23 October 2019, at NISSEI BILIS Co., Ltd., Shiga laboratory (Shiga, Japan) in accordance with Good Laboratory Practice regulations (https://www.mhlw.go.jp/). The results were evaluated by the Kureha Special Laboratory Co., Ltd. (Tokyo, Japan) as a third-party assessment to eliminate the researchers’ biases. The primary endpoint of this study was the safety, and the secondary endpoint was the efficacy of a PLGA/β-TCP for alveolar ridge preservation.

### 2.1. Animals

Eight male beagle dogs, aged 12–13 months old, weighing 9–13 kg were used. The dogs were brought to the facility to get used to the circumstances for more than one week before the surgery; were housed with a cage made of stainless steel individually during the experiment period in a temperature (18 °C–28 °C) and humidity (30%–80%)-controlled room, with an inverse 12 h day-night cycle with lights on at 7:00 and off at 19:00; and were fed with a daily soft pellet diet and water.

### 2.2. Experimental Design and Surgical Procedures

Anesthesia was induced by intravenous administration of thiamylal sodium (22.5 mg/kg) and intramuscular injection of butorphanol tartrate (0.1 mg/kg) followed by inhalation of an oxygen–isoflurane mixture (0.5%–3.0%). The mandibles were disinfected with application of 10% povidone-iodine, and the surgical region was locally anesthetized by lidocaine/adrenaline bitartrate. Post-operatively, 100 mg/body/day of ampicillin sodium was administered for 3 days. For post-operative pain, meloxicam (0.2 mg/kg) was injected subcutaneously at post-operative day 1. On both sides of the mandible in all dogs, the distal roots of premolar 3 (P3) and premolar 4 (P4) were extracted without raising a flap. The pulp tissue of these mesial roots was extirpated, and the root canals were filled with calcium hydroxide and iodoform paste (Vitapex, Neo Dental co., Tokyo, Japan) and sealed with polycarboxylate cement (HY-Bond Temporary Cement Soft, SHOFU INC., Kyoto, Japan). A total number of 32 extraction sites were prepared, four extraction sites in each dog. Extraction sites were assigned to one of the following treatments: (1) PLGA/β-TCP, (2) β-TCP (Cerasorb M, granule size 500–1000 µm, Curasan AG, Kleinostheim, Germany), one of the most popular commercially available β-TCP as the comparative experimental material, or (3) left empty as a control arranged in a sequence. Namely, the right P3 extraction socket was left empty, PLGA/β-TCP was implanted in the right P4 extraction socket, β-TCP was in the left P3 extraction socket, and the left P4 extraction was left empty in the first dog. In the second dog, PLGA/β-TCP was implanted in the right P3 extraction socket, β-TCP was in the right P4 extraction socket, the left P3 extraction socket was left empty, and PLGA/β-TCP was in the left P4 extraction socket, and so on. Consequently, 5 dogs (20 extraction sites, PLGA/β-TCP: n = 6, β-TCP: n = 6, control: n = 8) for 12 weeks evaluation and 3 dogs (12 extraction sites, PLGA/β-TCP: n = 4, β-TCP: n = 4, control: n = 4) for 24 weeks evaluation were prepared. The sample size was determined with reference to similar previous studies to use bigger experimental animals such as dog and sheep [14,15]. We examined more samples at 12 weeks than at 24 weeks, because previous studies showed β-TCP was almost resorbed and wound healing became stable at 24 weeks while wound healing was progressing at 12 weeks in alveolar ridge preservation. Each bone substitute was placed into the socket to the level of crestal bone and adapted to the internal socket morphology, with care given not to overfill the socket. Keratinized mucous tissue of 5 mm in diameter was harvested from the palate and was transplanted and sutured to close all extraction sockets. Five dogs at 12 weeks and 3 dogs at 24 weeks after surgery were euthanized using an overdose of pentobarbital (60 mg/kg). The mandibles were dissected, and each specimen was harvested.

The β-TCP of both the experimental groups have the same composition of Ca_3_(PO_4_)_2_, over 99% purity, 500–1000 µm granule size, and similar porosity of ~65%, but different pore size, i.e., Cerasorb M; 5–500 µm vs. GUIDOR *easy-graft*; 1–10 µm and also surface structure.

### 2.3. Observation of Wound Healing

Wound healing was observed at 1, 2, 4, 12, and 24 weeks post-operatively and scored with the following points: Complete wound dehiscence, 0 point; partial wound dehiscence or part of material exposure, 1 point; complete wound closure, 2 points. Three measures assessed the score, and the mean value was presented.

### 2.4. Radiographic Analysis

The specimens were scanned using a micro-computed tomography (µ-CT) system (Scan Xmate-L090, Comscantecno Co., Ltd., Yokohama, Japan) under standardized conditions, voltage 80 kV, current 100 µA, resolution 70.970 µm/pixel, before embedding in glycol methacrylate. The scanned data were reconstructed to three- and two-dimensional images using an image-analysis software (TRI/3D-BON, Ratoc System Engineering Co., Ltd., Tokyo, Japan). Following µ-CT analysis, a region of interest (ROI) bearing the same size as the original bone defect (5 × 3 × 3 mm [height × width × length]) was identified, and its position was determined in each of the axial, sagittal, and coronal planes. The horizontal position of the ROI was determined at the center of the extraction sockets, and its vertical position was determined by the lingual alveolar bone crest. The volume of mineralized tissue including bone tissue and residual granules in the ROI was defined as bone volume (BV).

### 2.5. Histological Analysis

The specimens were prepared using standard procedures for histological analysis. Briefly, the specimens were fixed with 10% formalin neutral buffer solution, degreased, dehydrated in an ascending series of alcohol solutions, and finally embedded in a glycol methacrylate resin. After polymerization, the specimens were sectioned in 6 µm thicknesses along the longitudinal axis of the extraction socket center with a fully motorized rotary microtome (RM2255, Leica, Nussloch, Germany). The slides were stained with toluidine blue and hematoxylin and eosin (HE), and were observed in a normal-light microscope (BX-51, Olympus, Tokyo, Japan). Light micrographs of the sections stained with toluidine blue were used for histomorphometric measurements. The percentage of newly formed bone in the center of the extraction sockets measurement region (8.640 mm^2^), which was determined on the basis of the lingual alveolar bone crest to 4.8 mm in depth, was calculated as the area of newly formed bone (n-Bone). Measurement of these histologic parameters was performed with reference to the previous study with minor modification [15]. Similarly, the percentage of remaining graft particles in the defect was calculated as the area of remaining biomaterial (r-Biomaterial). These areas were quantified on a computer using the NIH Image-J medical imaging software. The wound healing state and inflammatory reaction were evaluated by the sections stained with HE.

### 2.6. Evaluation of Operability of the Graft Materials

To compare the operability of PLGA/β-TCP and β-TCP, a mock study of alveolar ridge preservation was performed using a phantom (Figure 1A). Ten subjects participated in this experiment. Before application, each graft material was prepared according to usage instructions. The subjects received a transplant of each graft material into the extraction socket of the right maxillary second premolar (Figure 1B), which was attached to a dental simulator in the horizontal position, and the time required for implantation was recorded. An examiner measured the time required from the beginning of preparation to the termination of grafting of the materials into the socket to the level of crestal bone with a stopwatch.

The graft materials were placed into the extraction socket of the right second premolar of a jaw model mounted in a head phantom.

### 2.7. Statistical Analysis

Differences between treatment periods within groups were analyzed by a Wilcoxon signed-rank test. Statistical comparisons between different treatment groups were performed by a Kruskal–Wallis test with Dunn’s multiple comparisons test. These analyses were performed using a software package (JMP^®^ Pro13.0, SAS Institute Inc., Cary, NC, USA). Statistical significance was set at *p* < 0.05.

## 3. Results

### 3.1. Evaluation of Wound Healing

Macroscopically, there were no differences in soft tissue healing among each group; no severe inflammatory reactions or adverse reactions were observed in either experimental groups. Figure 1 shows the pre-operative and post-operative wound healing cascade images. In the test and control groups, gingival epithelization progressed over time and there were no wound healing complications nor surgical site infections during the follow-up period. The post-operative wound healing scores are shown in Table 1. Protracted wound healing was observed with PLGA/β-TCP treatment compared to that of β-TCP. Dehiscence of the wounded area was not found, though a small portion of the grafted material was exposed at 1 and 2 weeks post-operative with PLGA/β-TCP treatment (white arrowhead in Figure 2).

The gingival epithelization progressed over time, and no wound healing complications nor surgical site infections were observed in any groups. A small portion of the grafted material was exposed at 1 and 2 weeks post-operative in PLGA/β-TCP (white arrowhead). PO indicates post-operative weeks.

### 3.2. Radiographic Analysis

The volume of mineralized tissue including bone and residual granules in the ROI was quantitatively measured as BV by µ-CT analysis (Table 2). At 12 weeks, the PLGA/β-TCP and β-TCP values were significantly different compared to those of the control sites (Kruskal–Wallis test). The BV of β-TCP was also significantly higher compared to that of PLGA/β-TCP. At 24 weeks, there was no significant difference among the groups at this stage.

### 3.3. Histological Analysis

The amount of n-Bone and r-Biomaterial was quantitatively measured by histomorphometric analysis using slides stained with toluidine blue (Figure 3); a ROI is shown with a square in Figure 4a–c. At 12 weeks, the mean n-Bone of control, PLGA/β-TCP, and β-TCP was 31.4 ± 6.6%, 29.4 ± 20.4%, and 49.4 ± 7.6%, respectively. A significant difference was observed between β-TCP and the other groups. At 24 weeks, the mean n-Bone of control, PLGA/β-TCP, and β-TCP was 30.8 ± 4.9%, 33.8 ± 1.6%, and 38.0 ± 8.9%, respectively, with no significant difference among the groups. At 12 weeks, the mean r-Biomaterial of PLGA/β-TCP was significantly higher than that of β-TCP at 16.2 ± 10.7% and 3.7 ± 2.5%, respectively. At 24 weeks, the mean r-Biomaterial of PLGA/β-TCP and β-TCP was 0.2 ± 0.2% and 2.1 ± 1.2%, respectively, without any significant difference between the groups. As a significant difference in n-Bone and r-Biomaterial was observed mainly at 12 weeks, the wound healing state and inflammatory reaction were evaluated using HE-stained sections. Toluidine blue staining (Figure 4a–c) and HE (Figure 4d–f)-stained sections at post-operative 12 weeks show that new bone formation (* in Figure 4a–f) was evident in the extraction sockets, and these were observed well around bone substitutes (black arrowheads in Figure 4b, c). However, the bone substitutes appeared almost fragmented in the β-TCP group (dashed line in Figure 4f) while granules of 100–500 μm remained in the PLGA/β-TCP group (dashed line in Figure 4e). PLGA/β-TCP granules could be confirmed at the upper part of the extraction socket (white arrowheads in Figure 4b). Foreign body giant cells and the inflammatory cells were observed around PLGA/β-TCP and β-TCP (arrows in Figure 5), and the foreign body reaction tended to be strong around PLGA/β-TCP.

Histomorphometric findings regarding n-Bone and r-Biomaterial are shown. At 12 weeks, a significant difference in n-Bone was observed between β-TCP and the other groups. A significant difference in r-Biomaterial was observed between PLGA/β-TCP and β-TCP groups. At 24 weeks, there was no significant difference in n-Bone and r-Biomaterial area. * *p* < 0.05, significant difference.

Toluidine blue (a–c) and HE (d–f)-stained sections with high-power magnification at post-operative 12 weeks. An ROI is shown with a square in a–c. New bone formation (* in a–f) in extraction sockets was evident and was observed around bone substitutes (arrowheads in b, c). Whereas the bone substitutes in the β-TCP group were observed to be almost fragmented (dashed line in f), granules of 100–500 μm remained in the PLGA/β-TCP group (dashed line in e). PLGA/β-TCP granules could be confirmed at the upper part of the extraction socket (white arrowheads in b).

Foreign body giant cells and the inflammatory cells were observed around PLGA/β-TCP and β-TCP (arrows), and the foreign body reaction tended to be strong around PLGA/β-TCP.

### 3.4. Evaluation of Operability

The mean implantation time required of β-TCP (11.7 ± 53.8 s) was almost three times more than that of PLGA/β-TCP (41.2 ± 17.8 s) with a statistically significant difference (Figure 6).

## 4. Discussion

We assessed the safety and efficacy of PLGA/β-TCP for alveolar ridge preservation in extraction sockets compared to those of the conventional particulate β-TCP granules. PLGA/β-TCP and β-TCP scaffolds supported new bone formation without any notable adverse events. However, bone formation and substrate resorption were faster in the β-TCP group compared to those in the PLGA/β-TCP group in the early stage of bone regeneration, although comparable bone formation was observed in the PLGA/β-TCP group at the later stage versus that seen in the β-TCP group.

The gingival healing score tended to be lower in the PLGA/β-TCP group compared to that in the β-TCP group (Table 1). This may be due to the small portion of PLGA/β-TCP particles that was visible during the primary wound healing in this group (Figure 2). We assume that this was caused by the physical character of PLGA/β-TCP rather than any toxic effect. PLGA binds β-TCP granules together and immobilizes them, although a projecting granule may be exposed during the soft tissue healing process. Conversely, a β-TCP granule of the β-TCP group would be exhausted spontaneously if it protruded from the extraction socket.

In terms of bone regeneration, the amounts of mineralized bone and residual granules within the ROI for PLGA/β-TCP and β-TCP groups were significantly different compared to those of the control group at 12 weeks, though there were no significant differences between the treatment groups at 24 weeks. In addition, there was a decrease in BV for both PLGA/β-TCP and β-TCP groups in the latter stage of the study (Table 2). This phenomenon was notably present in the β-TCP group, suggesting that absorption of the β-TCP grafts occurs earlier than that of the PLGA/β-TCP grafts. The β-TCP group contained significantly more newly formed bone area than the PLGA/β-TCP group in comparison to that of the control group at 12 weeks, which was coincident with a significantly higher amount of r-Biomaterial in the PLGA/β-TCP group at the same stage (Figure 3). Ultimately, this delay in bone regeneration in the PLGA/β-TCP group at 12 weeks could be due to the slow degradation rate of PLGA/β-TCP. Although we cannot account for the exact reason for this delay in bone formation in the PLGA/β-TCP group, there are two possible reasons. It is possible that PLGA, an additive for PLGA/β-TCP, caused this phenomenon, and it is also possible that a different nature of the β-TCP used in both the experimental groups caused the difference.

PLGA is a biodegradable scaffold used for tissue healing; it degrades by hydrolysis within a few weeks and is known to be biocompatible [16]. Various materials coated with PLGA are widely used as a bone substitute, although it has been speculated that the PLGA coating may inhibit bone regeneration at graft sites [14]. Grafted PLGA is known to cause inflammatory responses due to an accumulation of lactic acid and glycolic acid produced by hydrolysis [17]. Accordingly, we observed foreign body giant cells and inflammatory cells around PLGA/β-TCP (Figure 5). The NMP used as a solvent for PLGA may also have caused the delay in bone regeneration in the PLGA/β-TCP group, as NMP is reported to induce a systemic effect that causes a decrease in body weight and food consumption without acute toxicity at the local site [18]. However, NMP and its resorption products are excreted primarily through urine in 1–3 days [19], and a single administration of NMP would have little cellular toxicity because of this high rate of metabolism. Moreover, NMP is flushed out upon contact with body fluids after the application of PLGA/β-TCP, suggesting that NMP may have a short-term cytotoxic effect [20], and would therefore not affect bone regeneration at 12 weeks post-operatively. Furthermore, preclinical and clinical studies comparing PLGA-coated biphasic calcium phosphate granules to the identical uncoated biphasic calcium phosphate granules showed no negative effects of the PLGA coating nor the addition of NMP [13,14]. This result suggests that the coating is unlikely the cause of the delayed bone formation.

In terms of β-TCP used in this study, it was better to use the identical β-TCP if the objective of the present study was the analysis of the effect of PLGA and/or NMP on bone regeneration. However, it was already studied in the previous study referred above [13,14]. Then, we thought it was reasonable to evaluate the safety and efficacy of PLGA/β-TCP for alveolar ridge preservation in extraction sockets compared to an already authorized material, because we aimed to get marketing authorization of the PLGA/β-TCP. The β-TCPs used in this study have the very similar property but are different in pore size and surface structure. Therefore, it is possible that a different nature of β-TCP caused the delay of bone formation in the PLGA/β-TCP group. Nevertheless, we assume that it is most likely that the slower degradation of PLGA/β-TCP caused the delay of bone formation and the appearance of more multinuclear giant cells in the PLGA/β-TCP group at 12 weeks post-operatively, because it is reported that active multinuclear cells appear on the surface of β-TCP to remove calcium and phosphate ions during the degradation of the material [21]. Figure 4 shows that resorption of the material was still progressing around PLGA/β-TCP, whereas conventional β-TCP was almost resorbed in 12 weeks. This progression of the resorption may explain the larger standard deviation of the PLGA/β-TCP group at 12 weeks post-operatively in Figure 3.

Bone regeneration was similar in the extraction sockets at 24 weeks post-operatively, although bone regeneration was evident in β-TCP at 12 weeks. The benefits of PLGA/β-TCP were only weakly apparent in the present experimental model because of the following reasons. First, the bone defects of the four-wall extraction sockets have high bone regenerative potential, so regeneration was completed without any alveolar ridge resorption even in a control. However, alveolar ridge preservation procedures have been shown to be effective in minimizing both horizontal and vertical post-extraction hard tissue dimensional loss [22]. Additionally, damaged extraction sockets benefit more from alveolar ridge preservation procedures compared to extraction sockets with intact walls [22]. A previous study comparing the potential for ridge preservation of PLGA/β-TCP and conventional particulate β-TCP using a dog extraction sockets model reported a significantly greater bone volume in PLGA/β-TCP graft sites at 12 weeks [20]. Although the grafts were gradually resorbed and replaced with newly formed and mature bone, in the β-TCP/PLGA group, some particles were encapsulated in connective tissue because of delayed dissolution and resorption of PLGA. In the present study, a significant higher amount of new bone formation was observed in the β-TCP group compared to the PLGA/β-TCP group at 12 weeks. This was because of the difference in the extraction sockets model with or without buccal bone deficiency. PLGA/β-TCP might be useful for buccal bone deficiency because of the formability and slower resorption rate. Next, β-TCP granules may have been scattered from extraction sockets unless the socket was covered with mucosal membrane. The PLGA/β-TCP graft is useful for buccal bone deficiency or vertical alveolar bone reduction because of the formability and slower resorption rate. Therefore, the addition of PLGA to β-TCP may result in a stable, solid alloplastic bone substitute, preventing the loss of exposed granules and also serving as a barrier membrane blocking soft tissue ingrowth [10] In contrast, Saito et al. reported that PLGA/β-TCP does not function as a barrier membrane, partitioning the soft tissue and grafted socket, because the latter might have contributed to the poorer bone quality [12]. Further studies are necessary to determine the efficacy of the β-TCP/PLGA bone substitute in ridge preservation.

The present study was designed to evaluate the safety of PLGA/β-TCP with an in vivo study in the dog tooth extraction model covered with mucous tissue to close all extraction sockets. We demonstrated that there were no significant adverse reactions against PLGA/β-TCP. The progression of wound healing in both soft and hard tissue was comparable to that of the empty control and conventional β-TCP. Although there was a delay in bone regeneration at the early stage of bone healing using the PLGA/β-TCP graft, the comprehensive results were favorable. Furthermore, we performed a mock study of alveolar ridge preservation using a phantom (Figure 1). This took approximately one third of the time to implant the PLGA/β-TCP graft into the extraction socket of the maxillary second premolar compared to that taken for a conventional β-TCP graft (Figure 6). Therefore, we concluded that the PLGA and the NMP-containing plasticizer that facilitate the in situ hardening properties of the material have no negative influence on the biocompatibility of β-TCP, and it may be beneficial to use PLGA/β-TCP for alveolar bone regeneration that requires restoration of the 3D configuration.

## Figures and Tables

**Figure 1 materials-13-03452-f001:**
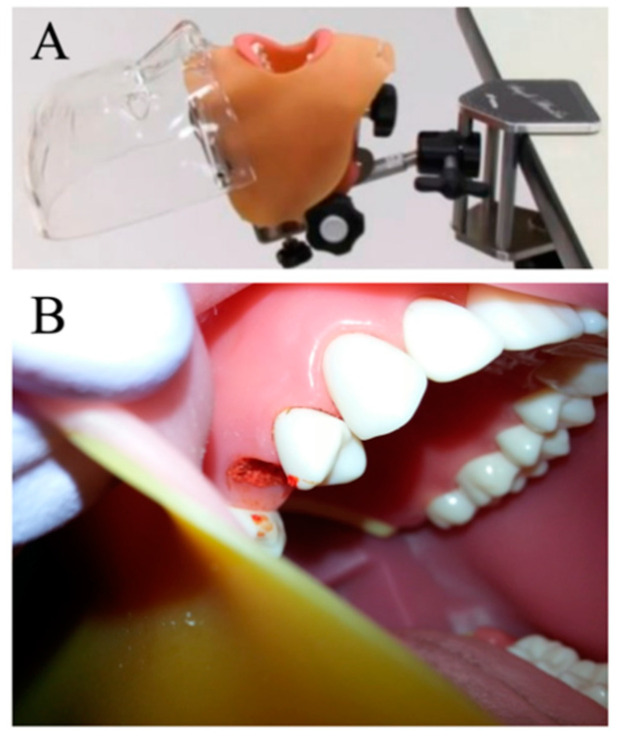
(**A**) phantom; (**B**) the extraction socket of the right maxillary second premolar

**Figure 2 materials-13-03452-f002:**
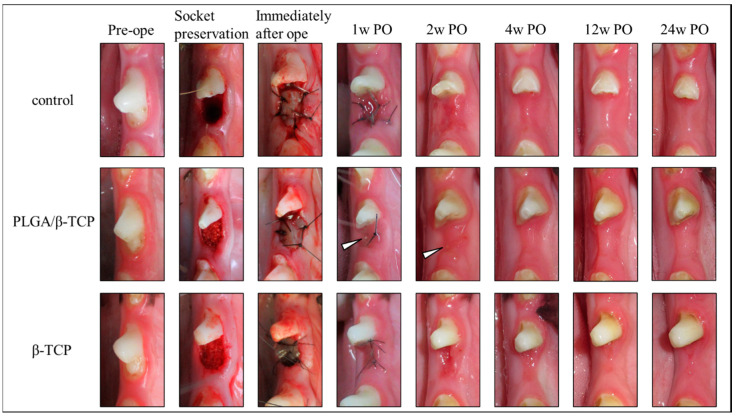
Pre-operative images and post-operative wound healing cascade.

**Figure 3 materials-13-03452-f003:**
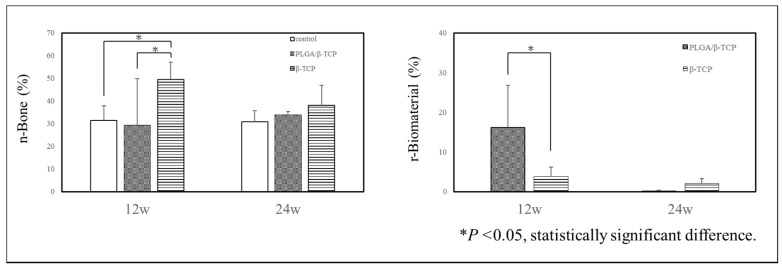
Quantitative histomorphometric analysis.

**Figure 4 materials-13-03452-f004:**
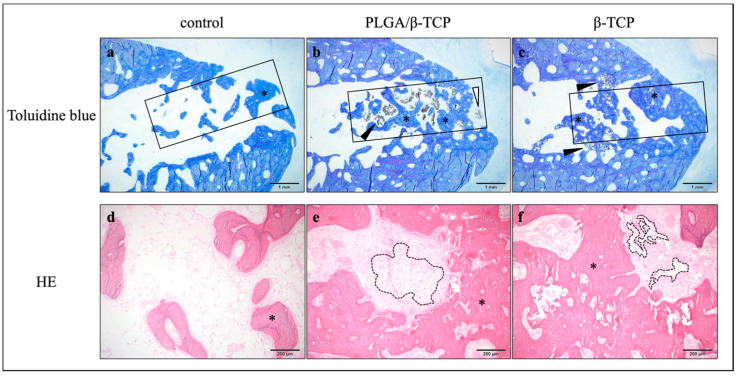
Histological analysis at 12 weeks post-operative. (**a**–**c**) toluidine blue stained sections; (**d**–**f**) HE stained sections.

**Figure 5 materials-13-03452-f005:**
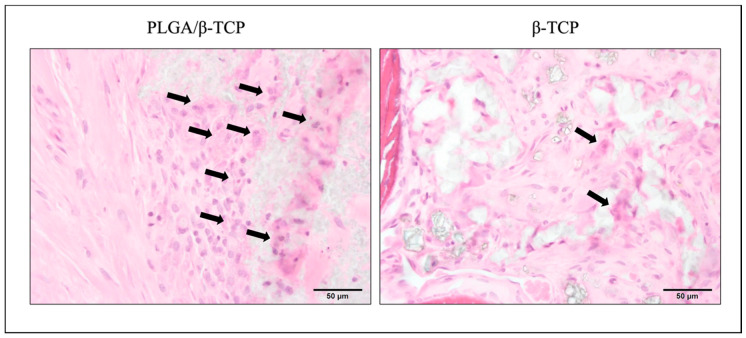
Higher magnification around bone substitute at 12 weeks post-operative.

**Figure 6 materials-13-03452-f006:**
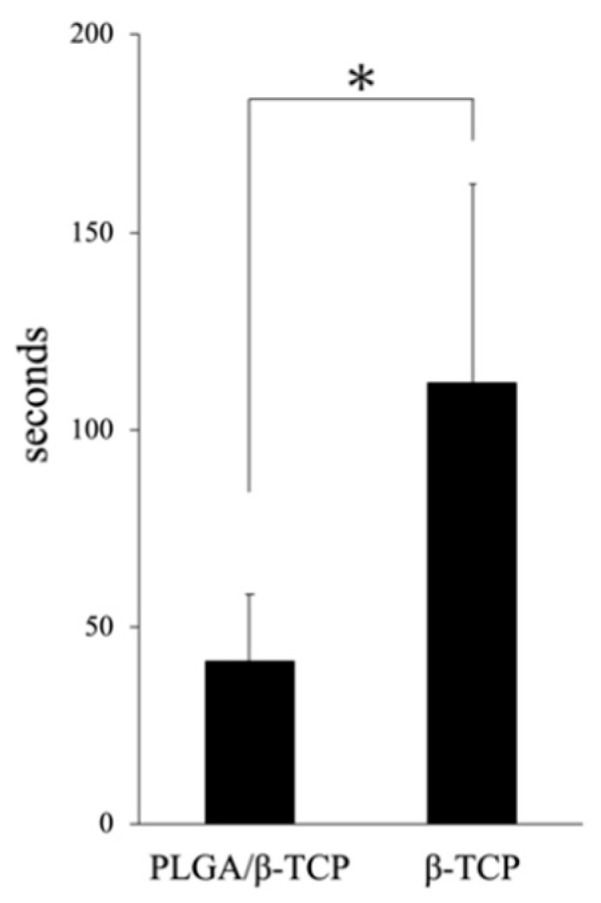
The time required for material graft. *: *p* < 0.05, significant difference.

**Table 1 materials-13-03452-t001:** Post-operative wound healing score.

		1w PO	2w PO	4w PO	12w PO	24w PO
**Control**	1~12w PO (n = 12) 24w PO (n = 4)	1.9 (0.3)	2.0 (0.0)	2.0 (0.0)	2.0 (0.0)	2.0 (0.0)
**PLGA/** **β-TCP**	1~12w PO (n = 10) 24w PO (n = 4)	1.2 (0.4) *	1.5 (0.5) *	2.0 (0.0)	2.0 (0.0)	2.0 (0.0)
**β-TCP**	1~12w PO (n = 10) 24w PO (n = 4)	1.8 (0.4)	1.9 (0.3)	2.0 (0.0)	2.0 (0.0)	2.0 (0.0)

Data are presented as Mean (SD); * *p* < 0.05, statistically significant difference to control and β-TCP; PO indicates post-operative weeks.

**Table 2 materials-13-03452-t002:** CT measurement of bone volume between 12 weeks and 24 weeks post-operative.

Radiographic Parameters	Post-operative	Control	PLGA/β-TCP	β-TCP
BV (mm^3^)	12w	19.8 (1.7)	23.1 (2.6) *	29.4 (1.9) * †
	24w	20.9 (4.9)	20.9 (2.6)	23.5 (3.4)

Data are presented as Mean (SD); * *p* < 0.05, statistically significant difference to control; † *p* < 0.05, statistically significant difference to PLGA/β-TCP.

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
