# Peer review of "Evaluation of Poly Lactic-co-Glycolic Acid-Coated β-Tricalcium Phosphate Bone Substitute as a Graft Material for Ridge Preservation after Tooth Extraction in Dog Mandible: A Comparative Study with Conventional β-Tricalcium Phosphate Granules"

_materials, 2020, doi:10.3390/ma13163452_

Round 1
Reviewer 1 Report
-The materials implanted were scaffolds or granules? In abstract authors used the word “scaffolds”: “Micro-CT images and histological observation revealed that both PLGA/β-TCP and β-TCP scaffolds supported sufficient new bone formation. However, sometimes along the manuscript the term “granules” is also used. Page 5, Line 177: “PLGA/β-TCP granules could be confirmed at the upper part of extraction socket …”
-Were both the materials commercially acquired? Information regarding the two materials must be presented (such as main properties, composition, granule size, etc).
-Are both materials FDA approved?
-Table 1 and Table 2 present (exactly) the same information. Based in the manuscript, data presented in Table 2 need to be changed.
-It is possible to find works that already evaluated the capacity of PLGA/β-TCP for ridge preservation after tooth extraction. Clin Adv Periodontics, 2017; 7(4): 190-194. DOI: 10.1902/cap.2017.160092. Please discuss this and explain the main differences between the work submitted and others already published.
Author Response
We appreciate your valuable comments and suggestions. Following those suggestions, we revised our manuscript with best of our efforts. Your comments were highly insightful and enabled us to improve the quality of our manuscript. Our point-by-point responses to each of your comments are the followings.
The materials implanted were scaffolds or granules? In abstract authors used the word “scaffolds”: “Micro-CT images and histological observation revealed that both PLGA/β-TCP and β-TCP scaffolds supported sufficient new bone formation. However, sometimes along the manuscript the term “granules” is also used. Page 5, Line 177: “PLGA/β-TCP granules could be confirmed at the upper part of extraction socket …”
Thank you for your indication. We used the term “scaffold” as a substrate for bone formation, but this expression leads to misunderstanding. Then, we revised “scaffold” to “granules”
-Were both the materials commercially acquired? Information regarding the two materials must be presented (such as main properties, composition, granule size, etc).
Yes, it is an important point. We explained it in the Introduction as “In addition, this material is commercially available widely in Europe and North America but not in Asia except Thailand and Singapore. Then, the present study was conducted as a part of non-clinical quality test to get marketing authorization.”, at line 66-69. We have also added the following sentence in the Discussion, “In terms of β-TCP used in this study, we used one of the most popular commercially available β-TCP (Cerasorb M) as the comparative experimental material, which is approved for marketing authorization widely in Asia including Japan.”, at line 288-290.
In terms of the property of the both materials, we added the explanation in the Discussion as, “The β-TCP of both the experimental groups have the same composition of Ca3(PO4)2, over 99% purity, 500 ~ 1000 µm granule size, and similar porosity of ~65%, but the different pore size, i.e. Cerasorb M; 5 ~ 500 µm vs. GUIDOR easy-graft; 1 ~10 µm and also surface structure.”, at line 295-298.
-Are both materials FDA approved?
Yes, they are.
-Table 1 and Table 2 present (exactly) the same information. Based in the manuscript, data presented in Table 2 need to be changed.
Thank you for the indication. We uploaded the wrong data. We replaced it to a correct one.
-It is possible to find works that already evaluated the capacity of PLGA/β-TCP for ridge preservation after tooth extraction. Clin Adv Periodontics, 2017; 7(4): 190-194. DOI: 10.1902/cap.2017.160092. Please discuss this and explain the main differences between the work submitted and others already published.
Thank you for your suggestion. We referred this paper and added to the Discussion as the following, “Whereas Saito et al. reported that PLGA/β-TCP does not function as a barrier membrane, partitioning the soft tissue and grafted socket, because the latter might have contributed to the poorer bone quality.12 Further studies are necessary to determine the efficacy of the β‐TCP/PLGA bone substitute in ridge preservation.”, at line328-331.
Reviewer 2 Report
Thank you for a very thorough and well-written study. I have no comments or remarks but my suggestions are as follows:
1. Line 86: "A total number of 32 extraction sites was prepared" - please replace "was" with "were" since it refers to "sites" and not "number". Thank you
2. There was no mention in the discussion on why you think there was a stronger foreign body reaction to the PLGA/b-TCP group. Why do you think that is?
3. It would be more impactful if you added in your introduction and discussion some articles that attempted to use the material in in-vitro and in-vivo and show how your results were similar/different
Author Response
We appreciate your valuable comments and suggestions. Following those suggestions, we revised our manuscript with best of our efforts. Your comments were highly insightful and enabled us to improve the quality of our manuscript. Our point-by-point responses to each of your comments are the followings.
- Line 86: "A total number of 32 extraction sites was prepared" - please replace "was" with "were" since it refers to "sites" and not "number".
Thank you for the comment. We have revised according to a reviewer’s comment.
- There was no mention in the discussion on why you think there was a stronger foreign body reaction to the PLGA/b-TCP group. Why do you think that is?
It is possible that PGLA, the coating material, caused this phenomenon. Then, we have already discussed this point and described as “Grafted PLGA is known to cause inflammatory responses due to an accumulation of lactic acid and glycolic acid produced by hydrolysis.16 Accordingly, we observed foreign body giant cells and inflammatory cells around PLGA/β-TCP (Figure 5).” at line 273-275. However, the previous study showed these coating materials, PLGA and NMP, had no negative effects on bone formation. Alternatively, β-TCP itself induces multinuclear giant cells during its decay. Thus, we assume the delay of resorption of PLGA/β-TCP may affect the appearance of phagocytotic cells and the resorption of this material was still progressing around PLGA/β-TCP, whereas conventional β-TCP was almost resorbed in 12 weeks. Consequently, we added the following sentence to the Discussion, “Nevertheless, we assume that it is most likely that the slower degradation of PLGA/β-TCP caused the delay of bone formation and the appearance of more multinuclear giant cells in PLGA/β-TCP group at 12 weeks post-operatively, because it is reported that active multinuclear cells appear on the surface of β-TCP to remove calcium and phosphate ions during the degradation of the material.21”, at line 299-303.
- It would be more impactful if you added in your introduction and discussion some articles that attempted to use the material in in-vitro and in-vivo and show how your results were similar/different.
Thank you for your suggestion. We have already cited some articles as references to introduce PLGA/β-TCP in the Introduction at line 60-64, but we added some articles to present the condition of PLGA/β-TCP among similar combination graft materials in the Introduction, at line 48-52.
We also conducted comparison with a previous study in the Discussion as “A previous study comparing the potential for ridge preservation of PLGA/β-TCP and conventional particulate β-TCP using a dog extraction sockets model reported a significant greater bone volume in PLGA/β-TCP graft sites at 12 weeks. Although the grafts were gradually resorbed and replaced with newly formed and mature bone, in the ?-TCP/PLGA group, some particles were encapsulated in connective tissue because of delayed dissolution and resorption of PLGA. In the present study, significant higher amount of new bone formation was observed in β-TCP group compared to PLGA/β-TCP group at 12 weeks. This was because of difference in extraction sockets model with or without buccal bone deficiency. PLGA/β-TCP might be useful for buccal bone deficiency because of the formability and slower resorption rate.”, at line 314-323.
Thank you again for your thoughtful suggestions. We believe the quality of our manuscript has become much improved.

Reviewer 3 Report
The paper presents a comparison of a commercial PLGA coated B-TCP with liquid activator (GUIDOR easy graft) to B-TCP granules in alveolar ridge preservation after tooth extraction in a canine model. Whilst the GUIDOR easy graft has been applied clinically the influence of the polymer and liquid activator has not been investigated in wound healing of the tooth extraction socket. In this respect the work demonstrates both novelty and clinical and scientific impact. The introduction is well written and clearly introduces the motivation for the study.
On the whole the scientific method appears sound although I am not clear as to why 8 animals were used with 4 extraction sites for each - 32 sites is not divisible by the 3 groups. How were the 2 timepoints taken into account when it came to this random allocation? Is there evidence that this n value is sufficient?
Although it appears that a thorough statistical analysis was carried out, it is very obvious that standard deviations are not included within tables or when quoting values within the bulk of the text. These should be included throughout in order that the reader may get a better impression of the nature of variation between the different groups. This is true for histological scores and as well as volumes of mineralised tissue, remnant implant.
There should be some justification as to why 5 dogs were sacrificed at week 12 and only 3 at week 24. Was the statistically significant differences observed at week 12 and not week 24 attributable to the reduced n value at 24 weeks?
I question whether the images in Figure 4 are sufficient to demonstrate a stronger foreign body reaction in the case of the PLGA presence, the difference does not appear clear.
Unfortunately whilst the results are interesting I am not convinced they fully support the points raised within the discussion. A weakly apparent benefit of the PLGA/ TCP over the TCP is suggested yet the results show no real evidence of this. In fact the converse, a slightly negative effect appears to be the case.
A result that the presence of the PLGA does not detrimentally effect the wound healing is a result in itself, however I would suggest the mock study be moved from the supplementary data to the bulk of the paper. In that way the paper can demonstrate that wound healing was not significantly effected by the addition of the polymer component and that the ease of implantation was significantly increased.
Author Response
We appreciate your valuable comments and suggestions. Following those suggestions, we revised our manuscript with best of our efforts. Your comments were highly insightful and enabled us to improve the quality of our manuscript. Our point-by-point responses to each of your comments are the followings.
The paper presents a comparison of a commercial PLGA coated B-TCP with liquid activator (GUIDOR easy graft) to B-TCP granules in alveolar ridge preservation after tooth extraction in a canine model. Whilst the GUIDOR easy graft has been applied clinically the influence of the polymer and liquid activator has not been investigated in wound healing of the tooth extraction socket. In this respect the work demonstrates both novelty and clinical and scientific impact. The introduction is well written and clearly introduces the motivation for the study.
Thank you for your kind comment.
On the whole the scientific method appears sound although I am not clear as to why 8 animals were used with 4 extraction sites for each - 32 sites is not divisible by the 3 groups. How were the 2 timepoints taken into account when it came to this random allocation? Is there evidence that this n value is sufficient?
Thank you for the indication. We have concretely explained how the materials were implanted as, “Extraction sites were assigned to one of the following treatments, 1) PLGA/β-TCP, 2) β-TCP (Cerasorb M, granule size 500–1000 µm, Curasan AG, Kleinostheim, Germany) or 3) left empty as a control arranging in a sequence. Namely, right P3 extraction socket was left empty, PLGA/β-TCP was implanted in right P4 extraction socket, β-TCP was in left P3 extraction socket, and left P4 extraction was left empty in a first dog. In a second dog, PLGA/β-TCP was implanted in right P3 extraction socket, β-TCP was in right P4 extraction socket, left P3 extraction socket was left empty, and PLGA/β-TCP was in left P4 extraction socket, and so on. Consequently, 5 dogs (20 extraction sites, PLGA/β-TCP: n = 6, β-TCP: n = 6, control: n = 8) for 12 weeks evaluation and 3 dogs (12 extraction sites, PLGA/β-TCP: n = 4, β-TCP: n = 4, control: n = 4) for 24 weeks evaluation were prepared. We examined more samples at 12 weeks than that at 24 weeks, because previous studies showed almost β-TCP was resorbed and wound healing became stable at 24 weeks while wound healing was progressing at 12 weeks in alveolar ridge preservation.”, at line 94-106 in the Materials and Methods 2.2.
Although it appears that a thorough statistical analysis was carried out, it is very obvious that standard deviations are not included within tables or when quoting values within the bulk of the text. These should be included throughout in order that the reader may get a better impression of the nature of variation between the different groups. This is true for histological scores and as well as volumes of mineralised tissue, remnant implant.
We have extensively revised Table 1 and 2, and also added SDs in the text body. Furthermore, SDs for histological scores were also added in in the text body.
There should be some justification as to why 5 dogs were sacrificed at week 12 and only 3 at week 24. Was the statistically significant differences observed at week 12 and not week 24 attributable to the reduced n value at 24 weeks?
We speculated that there were much deviations in tissue response to the materials since extraction socket wound healing was progressing at 12 weeks, and the wound healing would be already stable at 24 weeks, according to previous studies, so that we decided to evaluate more samples at 12 weeks than that at 24 weeks. We explained this at line 103-106 as, “We examined more samples at 12 weeks than that at 24 weeks, because previous studies showed almost β-TCP was resorbed and wound healing became stable at 24 weeks while wound healing was progressing at 12 weeks in alveolar ridge preservation.” It is also the reason why statistical difference was big at 12 weeks. Then, we discussed this as, “Figure 4 shows that resorption of the material was still progressing around PLGA/β-TCP, whereas conventional β-TCP was almost resorbed in 12 weeks. This progression of the resorption may explain the larger standard deviation of PLGA/β-TCP group at 12 weeks post-operatively in Figure 3.”, at line 303-306.
I question whether the images in Figure 4 are sufficient to demonstrate a stronger foreign body reaction in the case of the PLGA presence, the difference does not appear clear.
Thank you for the comment. We believe Figure 4 demonstrated more foreign body giant cells in PLGA/β-TCP than that in β-TCP, although we did not perform quantitative analysis. We think this phenomenon was caused by the delay of resorption of β-TCP in PLGA/β-TCP. Then, we discussed this in the Discussion as, “Nevertheless, we assume that it is most likely that the slower degradation of PLGA/β-TCP caused the delay of bone formation and the appearance of more multinuclear giant cells in PLGA/β-TCP group at 12 weeks post-operatively, because it is reported that active multinuclear cells appear on the surface of β-TCP to remove calcium and phosphate ions during the degradation of the material.21”, at line 299-303.
Unfortunately whilst the results are interesting I am not convinced they fully support the points raised within the discussion. A weakly apparent benefit of the PLGA/ TCP over the TCP is suggested yet the results show no real evidence of this. In fact the converse, a slightly negative effect appears to be the case.
A result that the presence of the PLGA does not detrimentally effect the wound healing is a result in itself, however I would suggest the mock study be moved from the supplementary data to the bulk of the paper. In that way the paper can demonstrate that wound healing was not significantly effected by the addition of the polymer component and that the ease of implantation was significantly increased.
Thank you for your very kind suggestion. We totally agree with your view. Unfortunately, the results of the present study had not shown the benefit of PLGA/β-TCP, so that we added the mock study as the supplemental data to present the superiority of PLGA/β-TCP in handling. According to your kind suggestion, we moved the data into the text body.
We appreciate again for your kind suggestions.

Reviewer 4 Report
This manuscript reports on evaluation of the safety and efficacy of a poly lactic-co-glycolic acid (PLGA) coated β-tricalcium phosphate (β-TCP) with N-methyl-2-pyrrolidone (NMP) liquid activator (PLGA/β-TCP) on alveolar ridge preservation after tooth extraction in dog mandible.
The topic is specific to a dentistry journal. The scope is narrow on studying one commercial brand and comparing the results with those of another random commercial brand and the impact is moderate. In order to be published in Materials MDPI the authors will have to justify the choice of commercial materials and focus more on the impact statement for being published in a general materials journal. Besides, the quality of the work can be improved through careful proofreading and control of the correct data tabulation. Data presented in Table 2 are oddly the same as that in Table 1, unacceptable.
In conclusion, the work could be accepted after major revision.
Some suggestion and detailed points to be addressed by the authors:
1. In order to rule out the negative contribution of PLGA and NMP, the same β-TCP should be used. This is because ‘The difference in the nature of β-TCP in the two experimental groups may have caused the delay of the early stage bone regeneration…’ was inconclusive. And this defeats the purpose of testing the safety and efficacy of PLGA and NMP modification. Have the authors communicated with the manufacturers about the source of β-TCP?
2. The Introduction does not provide sufficient background. Also the introduction of this specific brand of β-TCP is very abrupt with no transition. A paragraph should be added first on specifics of β-TCP, e.g. what other synthetic materials have been used to combine with β-TCP, then introduce coating with PLGA, and the effects, etc. The imapct should be stated here.
3. L58, ‘wound hearing’ spelling mistake.
4. How long were the dogs housed for before and after experiment?
5. What are the operating parameters used for microCT scanning? Such information should not be omitted.
6. What is the rational behind the quantified parameters in Section 2.5?
7. Table 2 data is the same as Table 1. Did the authors not check their tabulated data? Mean (SD) and p values should all be reported.
8. ‘Post-operative N weeks’ should be changed to n weeks postoperative throughout.
9. Fig. 2, the SDs of both parameters for PLGA/β-TCP at 12 w are exceptionally large. What is the reason? This should be addressed.
10. In Discussion, only Table 2 was referred to once. More results should be referred to and discussed before the general discussion.
11. What is the purpose of the Supplementary Materials? It should be at least mentioned in the main text.
Author Response
We appreciate your valuable comments and suggestions. Following those suggestions, we revised our manuscript with best of our efforts. Your comments were highly insightful and enabled us to improve the quality of our manuscript. Our point-by-point responses to each of your comments are the followings.
The topic is specific to a dentistry journal. The scope is narrow on studying one commercial brand and comparing the results with those of another random commercial brand and the impact is moderate. In order to be published in Materials MDPI the authors will have to justify the choice of commercial materials and focus more on the impact statement for being published in a general materials journal. Besides, the quality of the work can be improved through careful proofreading and control of the correct data tabulation. Data presented in Table 2 are oddly the same as that in Table 1, unacceptable.
Thank you for pointing out the most critical issue of our manuscript. As you indicated, we compared the commercially available materials in an animal experiment, but PLGA/β-TCP is not authorized for marketing approval in Japan. Then, we intended to conduct this study to get marketing authorization. We added the following sentences in the Introduction to introduce our purpose of this study, “In addition, this material is commercially available widely in Europe but not in Asia except Thailand and Singapore. Then, the present study was conducted as a part of non-clinical quality test to get marketing authorization.”, at line 66-69. We have also revised the Discussion intensively to explain the reason for the select of the comparative materials as shown in the response to #1.
- In order to rule out the negative contribution of PLGA and NMP, the same β-TCP should be used. This is because ‘The difference in the nature of β-TCP in the two experimental groups may have caused the delay of the early stage bone regeneration…’ was inconclusive. And this defeats the purpose of testing the safety and efficacy of PLGA and NMP modification. Have the authors communicated with the manufacturers about the source of β-TCP?
Thank you again for pointing out very much critical point of this study. We explained the reason why we selected Serasorb M as the comparative experimental material in the Discussion as the followings, “In terms of β-TCP used in this study, we used one of the most popular commercially available β-TCP (Cerasorb M) as the comparative experimental material, which is approved for marketing authorization widely in Asia including Japan. It was better to use the identical β-TCP if the objective of the present study was the analysis of the effect of PLGA and/or NMP on bone regeneration. However, it was already studied in the previous study referred in above. 12, 13 Then, we thought it was reasonable to evaluate the safety and efficacy of PLGA/β-TCP for alveolar ridge preservation in extraction sockets compared with an already authorized material, because we aimed to get marketing authorization of the PLGA/β-TCP.”, at line 288-295.
As you indicated, we agree to ‘The difference in the nature of β-TCP in the two experimental groups…’ was inconclusive. We have not concluded but itemized the possibility. We have revised the Dicussion about β-TCP as the followings, “The β-TCP of both the experimental groups have the same composition of Ca3(PO4)2, over 99% purity, 500 ~ 1000 µm granule size, and similar porosity of ~65%, but the different pore size, i.e. Cerasorb M; 5 ~ 500 µm vs. GUIDOR easy-graft; 1 ~10 µm and also surface structure. Therefore, it is possible that a different nature of β-TCP caused the delay of bone formation in PLGA/β-TCP group. However, we assume that it is most likely that the slower degradation of PLGA/β-TCP caused the delay of bone formation and the appearance of more multinuclear giant cells in PLGA/β-TCP group at 12 weeks post-operatively, because it is reported that active multinuclear cells appear on the surface of β-TCP to remove calcium and phosphate ions during the degradation of the material.21“, at line 295-303.
- The Introduction does not provide sufficient background. Also the introduction of this specific brand of β-TCP is very abrupt with no transition. A paragraph should be added first on specifics of β-TCP, e.g. what other synthetic materials have been used to combine with β-TCP, then introduce coating with PLGA, and the effects, etc. The impact should be stated here.
Thank you for your helpful suggestion. We have cited some studies as reference 4, 5, 6 to introduce the condition of PLGA/β-TCP among similar combination graft materials at line 48-52 as, “In recent years, several bone substitutes based on calcium phosphates, combined with different natural-based polymers, such as cellulose and its derivatives, hyaluronic acid and other polymers, have been marketed as medical devices.4, 5 Although these materials fulfill the requirements of injectability, filling complex-shaped bone defects and setting very firmly in situ, nevertheless lack osteoconductivity and degradability.6 ”
- L58, ‘wound hearing’ spelling mistake.
Thank you for the comment. We have revised according to a reviewer’s comment.
- How long were the dogs housed for before and after experiment?
The dogs were brought to the facility to get used to the circumstances more than one week before the surgery and were housed with a cage made of stainless steel individually during the experiment period. We have added this information in Section 2.1. line 78-80.
- What are the operating parameters used for microCT scanning? Such information should not be omitted.
Scan condition was voltage 80 kV, current 100 µA, resolution 70.970 μm/pixel. We have added this information in Section 2.4. at line 118-119.
- What is the rational behind the quantified parameters in Section 2.5?
Measurement of these histologic parameters was performed with reference to previous study with minor modification. We have cited the study by Barros et al. as reference 14 at line 138-139.
- de Barros, R. R. M.; Novaes Jr, A. B.; de Carvalho, J. P.; de Almeida, A. L. G., The effect of a flapless alveolar ridge preservation procedure with or without a xenograft on buccal bone crest remodeling compared by histomorphometric and microcomputed tomographic analysis. Clinical oral implants research 2017, 28 (8), 938-945.
- Table 2 data is the same as Table 1. Did the authors not check their tabulated data? Mean (SD) and p values should all be reported.
Thank you very much for helping us, wrong data was uploaded in Table 2. We have replaced with correct data. We have also extensively revised the Table 1 according to the reviewer’s instruction.
- ‘Post-operative N weeks’ should be changed to n weeks postoperative throughout.
We have extensively revised the manuscript according to the editor’s instruction.
- 2, the SDs of both parameters for PLGA/β-TCP at 12 w are exceptionally large. What is the reason? This should be addressed.
Thank you for an important suggestion. We considered this phenomenon is due to delay in bone formation and slow degradation rate of PLGA/β-TCP. Although pure β-TCP were gradually resorbed and replaced with newly formed and mature bone until 12 weeks post-surgery, PLGA coating is more likely to delay resorption rate. Thus, we have added the sentences at line 303-306 as, “Figure 4 shows that resorption of the material was still progressing around PLGA/β-TCP, whereas conventional β-TCP was almost resorbed in 12 weeks. This progression of the resorption may explain the larger standard deviation of PLGA/β-TCP group at 12 weeks post-operatively in Figure 3.”
- In Discussion, only Table 2 was referred to once. More results should be referred to and discussed before the general discussion.
Thank you for the suggestion, but we have already discussed about all the results in the Discussion. To make it clear we have added references in the part which we discuss about the figures or tables.
- What is the purpose of the Supplementary Materials? It should be at least mentioned in the main text.
The Supplementary Materials aims to indicate the operability of PLGA/β-TCP. We recognized that it was slightly different matter from the purpose of this study, and so we thought it might be better to delete the supplementary materials. However, in this case, only slightly negative effects of PLGA/β-TCP would be presented in this paper. Then, we added the mock study to present the benefit of PLGA/β-TCP. As Reviewer 3 suggested like as you, we moved this date in the main text.
Thank you again for your incisive criticisms. All of them are very much helpful to improve the quality of our manuscript.

Reviewer 5 Report
The submitted manuscript by Koga et al. evaluates the safety and efficacy of alveolar ridge preservation procedures using either beta-TCP or PLGA coated beta-TCP with N-methyl-2-pyrrolidone in eight dog mandibles. Post-operative wound healing, bone regeneration using micro-CT images, and histomorphometric analyses were evaluated.
Introduction:
Line 50: please delete “the first” and replace it by “a”. There are other bone grafting materials with similar properties.
Material and methods:
What randomization method was used?
Was a sample size calculation performed?
Each animal had four extraction sockets. Three surgical procedures were performed ( empty control / beta-TCP / PLGA-beta-TCP). How was the fourth extraction socket treated?
Lines 122-123: the authors describe the remaining beta-TCP particles (graft-material) as “implant”. The term “implant” is usually associated with dental implants and should, therefore, be replaced to avoid confusion. (E.g. Remaining graft particles, remaining biomaterial, remaining TCP granules, …)
Tables 1&2: The caption of the tables should contain the meaning of the abbreviation “POD”
Discussion / supplementary material :
The lines 264 – 281 report on another study in which the treatment time required to preserve the extraction sockets in a phantom was analyzed. These lines should be deleted as they are not related to the submitted study.
The authors could rather add information on the volume stability of the extraction sockets if these data are available.
Author Response
We appreciate your valuable comments and suggestions. Following those suggestions, we revised our manuscript with best of our efforts. Your comments were highly insightful and enabled us to improve the quality of our manuscript. Our point-by-point responses to each of your comments are the followings.
Introduction:
Line 50: please delete “the first” and replace it by “a”. There are other bone grafting materials with similar properties.
Thank you for the comment. We have revised according to a reviewer’s comment.
Material and methods:
What randomization method was used?
“Extraction sites were randomly asigned to …” may lead misunderstanding. Then, we concretely explained the way of the implantation as the following, “arranging in a sequence. Namely, right P3 extraction socket was left empty, PLGA/β-TCP was implanted in right P4 extraction socket, β-TCP was in left P3 extraction socket, and left P4 extraction was left empty in a first dog. In a second dog, PLGA/β-TCP was implanted in right P3 extraction socket, β-TCP was in right P4 extraction socket, left P3 extraction socket was left empty, and PLGA/β-TCP was in left P4 extraction socket, and so on. Consequently, 5 dogs (20 extraction sites, PLGA/β-TCP: n = 6, β-TCP: n = 6, control: n = 8) for 12 weeks evaluation and 3 dogs (12 extraction sites, PLGA/β-TCP: n = 4, β-TCP: n = 4, control: n = 4) for 24 weeks evaluation were prepared.”, at line 95-103.
Was a sample size calculation performed?
We haven’t done power analysis to calculate ample size calculation, because the comparative experimental groups expected to show a little difference, so that sample size will be huge. Then, the sample size was determined with reference to some of similar previous studies to use bigger experimental animals such as dog not like mouse or rats. Each animal had four extraction sockets. Three surgical procedures were performed (empty control / beta-TCP / PLGA-beta-TCP).
How was the fourth extraction socket treated?
It is the same answer with that to #1. The fourth extraction socket was also treated with one of the experimental treatments. As a result, 5 dogs (20 extraction sites, PLGA/β-TCP: n = 6, β-TCP: n = 6, control: n = 8) for 12 weeks evaluation group and 3 dogs (12 extraction sites, PLGA/β-TCP: n = 4, β-TCP: n = 4, control: n = 4) for 24 weeks evaluation group were created.
Lines 122-123: the authors describe the remaining beta-TCP particles (graft-material) as “implant”. The term “implant” is usually associated with dental implants and should, therefore, be replaced to avoid confusion. (E.g. Remaining graft particles, remaining biomaterial, remaining TCP granules, …)
Thank you for the helpful comment. We have revised “Implant” to “Biomaterial” in the manuscript and the figures according to reviewer’s comment.
Tables 1&2: The caption of the tables should contain the meaning of the abbreviation “POD”
We have revised Tables according to reviewer’s comment.
The lines 264 – 281 report on another study in which the treatment time required to preserve the extraction sockets in a phantom was analyzed. These lines should be deleted as they are not related to the submitted study.
We agree to this reviewer’s comment, and we thought to delete the supplementary materials. As suggested by the reviewer, it is weakness of this study. We had performed other analysis about the volume stability of the extraction sockets. For example, CT measurement of buccolingual alveolar bone width including the extraction socket and the measurement of vertical distance between the buccal and lingual crests. However, there were no significant difference in each group in all time point. It was thought to influence these results the bone defects of the four-wall extraction sockets have high bone regenerative potential, so that regeneration was completed without any alveolar ridge resorption even in a control as described in discussion. In this case, only slightly negative effects of PLGA/β-TCP would be presented in this paper. Reviewer 3 and 4 suggested us to move this date in the main text. Then, we moved the mock study data to the main text to present the benefit of PLGA/β-TCP. Thank you for your understanding.

Round 2
Reviewer 1 Report
No comments or suggestions.
Author Response
Thank you very much for your time to review our manuscript.
Reviewer 3 Report
Thank you for addressing the comments I raised, the manuscript is very much improved. There are places where small changes in grammer would be beneficial, but on the whole a much more conclusive piece of work is now presented.
Author Response
Thank you very much for your kind comments.
Reviewer 4 Report
The quality and scientific soundness have been improved after the first round of revision by the authors through careful and diligent addressing of nearly all aspects raised by the reviewers.
However, the following 5 points must be addressed before acceptance for publication.
1. One important point raised by this reviewer that the authors agreed upon, but did not implement '...focus more on the impact statement for being published in a general materials journal' must be addressed for justification of publication in Materials MDPI. Adding a new first paragraph consisting one to two sentences introducing general current state of bone grafting, history, rationale, selection of materials and technique citing ref [7], for example, before going into the specifics of implant dentistry, post-extraction alveolar ridge bone preservation in the second paragraph, should suffice to remedy this.
2. Another important class of non-absorbable grafting material glass ionomer cements must not be left out, please revise as such “Non-absorbable grafting material, such as hydroxyapatite and glass ionomer cements [a]...”
[a] Tian KV, Chass GA, Di Tommaso D, Simulations reveal the role of composition into the atomic-level flexibility of bioactive glass cements. Phys. Chem. Chem. Phys., 2016, 18, 837-845.
3. Section 3.2, repeating numerical results that have been tabulated in Table 2 is not needed.
4. L288-299 justification of choice of materials should be brought forward to where the materials were first introduced in Section 2.2.
5. Table 1, why are some SD 0.0? If this score is subject to personal opinion, the uncertainty should be addressed.
Author Response
Thank you for the comments. The followings are our point-by-point responses.
- One important point raised by this reviewer that the authors agreed upon, but did not implement '...focus more on the impact statement for being published in a general materials journal'must be addressed for justification of publication in Materials MDPI. Adding a new first paragraph consisting one to two sentences introducing general current state of bone grafting, history, rationale, selection of materials and technique citing ref [7], for example, before going into the specifics of implant dentistry, post-extraction alveolar ridge bone preservation in the second paragraph, should suffice to remedy this.
Thank you for your suggestion. We have added general information of bone grafting in introduction as “The recovery of bone deficiency caused by trauma, tumor resection, and also aging has been a challenge in the field of orthopedics and also dentistry. Autogenous bone graft is thought to be a gold standard because of its superior osteogenesity, however, it has impediments such as limited availability and a donor site morbidity. In order to compensate these drawbacks, variety of artificial bone graft materials has been developed and clinically applied for bone augmentation, and the materials selected significantly affect the outcome of bone replacement procedures in terms of bone formation volume and the quality and amount of vital bone.1”
- Another important class of non-absorbable grafting materialglass ionomer cementsmust not be left out, please revise as such “Non-absorbable grafting material, such as hydroxyapatite and glass ionomer cements [a]...”
[a] Tian KV, Chass GA, Di Tommaso D, Simulations reveal the role of composition into the atomic-level flexibility of bioactive glass cements. Phys. Chem. Chem. Phys., 2016, 18, 837-845.
Thank you for the comment. We have revised the manuscript according to a reviewer’s comment.
- Section 3.2, repeating numerical results that have been tabulated in Table2 is not needed.
Thank you for the comments. We have deleted the numerical results from section 3.2.
- L288-299 justification of choice of materialsshouldbe brought forward to where the materials were first introduced in Section 2.2.
Thank you for the comment. We have moved justification of choice of materials in section 2.2.
- Table 1, why are some SD 0.0?If this score is subject to personal opinion, the uncertainty should be addressed.
The wound healing score was assessed by three measurers, and the mean was presented. All measurers assessed the gingival score as 2 points after gingival epithelization was completed. Therefore, the SD had become 0.0. This information have added in section 2.3 as “Three measures assessed the score, and the mean value was presented.”

Reviewer 5 Report
The revised manuscript has been improved in many parts, but with the added passages new questions come up. Further changes are necessary.
In my opinion, the manuscript now focuses mainly on a commercial brand, promotes positive effects, and describes the local availability of study products in different countries. This is not the quality of an article that you would expect in a journal like “Materials”. Please adhere strictly to the study results.
Sentences such as „In addition, this material is commercially available widely in Europe and North America but not in Asia except Thailand and Singapore.“ (lines 66-68) should be deleted as they are not relevant. The brand names should only be mentioned once in the section “Materials and Methods” no further repetition is needed. Furthermore, information on availability may change over time. I am aware that another reviewer has asked questions in this direction, but the required and necessary information mainly concerned the question of whether the materials used are experimental or commercial products.
The information of the sentence “Then, the present study was conducted as a part of non-clinical quality test to get marketing authorization.” (line 68-69) contains new questions. First of all, the sentence does not belong here, but to the study design, so please delete it here, as well as the sentence before.
Most importantly, the question arises as to whether the study is sponsored by the company. Why do the authors "conduct non-clinical quality tests in order to obtain marketing authorization"? The main beneficiary of such studies is the manufacturing company, which normally pays for this type of study. When the payment is received, this should be mentioned in the conflict of interest section /funding section of the study.
Introduction
line 50: „...have been marketed as medical devices“ should be replaced/rephrased.
Materials and Methods
Sample size calculation is missing
Primary and secondary endpoints are missing
Results:
I still recommend that the passage “2.6 Evaluation of operability of the graft materials” should be removed. This data is not part of the conducted and described the study and seems to be driven by marketing ideas.
The same applies to lines 236-241 on the same topic.
Discussion
Again, I recommend removing all commercial brand names form the discussion section 283-285. Used products are mentioned only once in the M&M section.
Additionally, there are similar points already mentioned in the introduction section. Do not mention the availability of different products in different countries’. (lines 289-290).
Again the authors state “because we aimed to get marketing authorization” (line 294-295). This is basically a company driven study This information must be included in the conflict of interest or financing category of the study. But still, this cannot be the reason for conducting a study. Please stick to a scientific hypothesis.
Author Response
Thank you for your critical reading and suggestive comments. The followings are our point-by-point responses.
The revised manuscript has been improved in many parts, but with the added passages new questions come up. Further changes are necessary.
In my opinion, the manuscript now focuses mainly on a commercial brand, promotes positive effects, and describes the local availability of study products in different countries. This is not the quality of an article that you would expect in a journal like “Materials”. Please adhere strictly to the study results.
Sentences such as „In addition, this material is commercially available widely in Europe and North America but not in Asia except Thailand and Singapore.“ (lines 66-68) should be deleted as they are not relevant. The brand names should only be mentioned once in the section “Materials and Methods” no further repetition is needed. Furthermore, information on availability may change over time. I am aware that another reviewer has asked questions in this direction, but the required and necessary information mainly concerned the question of whether the materials used are experimental or commercial products.
We agree to a reviewer’s comment. We have deleted line 66-68 and the repetition of the brand name. As a reviewer indicated, we compared the commercially available materials in an animal experiment, but PLGA/β-TCP is not authorized for marketing approval in Japan. Then, we intended to conduct this study as a part of non-clinical quality test to get marketing authorization.
The information of the sentence “Then, the present study was conducted as a part of non-clinical quality test to get marketing authorization.” (line 68-69) contains new questions. First of all, the sentence does not belong here, but to the study design, so please delete it here, as well as the sentence before.
Thank you for the indication. We have deleted line 68-69.
Most importantly, the question arises as to whether the study is sponsored by the company. Why do the authors "conduct non-clinical quality tests in order to obtain marketing authorization"? The main beneficiary of such studies is the manufacturing company, which normally pays for this type of study. When the payment is received, this should be mentioned in the conflict of interest section /funding section of the study.
This study was conducted as collaborative research with the company.
Introduction
line 50: „...have been marketed as medical devices“ should be replaced/rephrased.
Thank you for the comments. We have revised the manuscript as “… have been used as medical devices.”
Materials and Methods
Sample size calculation is missing
We have added the information about the determination of sample size in section 2.2 as “The sample size was determined with reference to similar previous studies to use bigger experimental animals such as dog and sheep.14, 15”
Primary and secondary endpoints are missing
We have added the information about primary and secondary endpoints in the first paragraph of Materials and Methods as “The primary endpoint of this study was the safety, and the secondary endpoint was efficacy of PLGA/β-TCP for alveolar ridge preservation.”
Results:
I still recommend that the passage “2.6 Evaluation of operability of the graft materials” should be removed. This data is not part of the conducted and described the study and seems to be driven by marketing ideas.
We understand your strong opinion. However, we regard an operability as one of the important properties for bone substitute as describe in the Introduction. This study aimed to evaluate the efficacy of PLGA/bTCP as the secondary endpoint, so that evaluation of operability is an important item. The context of this experiment is slightly different from the main body of the text, but this item shows the unique characteristic of the examined material in this study. Then, we moved this part to the text body in accordance with the other reviewers’ suggestions.
The same applies to lines 236-241 on the same topic.
It is necessary for the reason same as above.
Discussion
Again, I recommend removing all commercial brand names form the discussion section 283-285. Used products are mentioned only once in the M&M section.
We have deleted the brand name in the Discussion.
Additionally, there are similar points already mentioned in the introduction section. Do not mention the availability of different products in different countries’. (lines 289-290).
We have deleted line 289-290.
Again the authors state “because we aimed to get marketing authorization” (line 294-295). This is basically a company driven study This information must be included in the conflict of interest or financing category of the study. But still, this cannot be the reason for conducting a study. Please stick to a scientific hypothesis.
As described above, this study was conducted as collaborative research with the company, so that we believe we do not need to present a source of the funding in COI. The suitability of PLGA/bTCP for ridge preservation has been clinically reported. However, the impact of polymers on the resorption/replacement of PLGA/β-TCP and negative effect of PLGA and NMP during wound healing is currently unknown. Furthermore, the detailed examination is not conducted about operability. Therefore, this study was conducted not only as a part of non-clinical quality test to get marketing authorization but also to evaluate the safety and efficacy of a PLGA/β-TCP.
We hope your kind understanding.
